# Alpha-Deoxyguanosine to Reshape the Alpha-Thrombin Binding Aptamer

**DOI:** 10.3390/ijms24098406

**Published:** 2023-05-07

**Authors:** Natalia A. Kolganova, Vladimir B. Tsvetkov, Andrey A. Stomakhin, Sergei A. Surzhikov, Edward N. Timofeev, Irina V. Varizhuk

**Affiliations:** 1Engelhardt Institute of Molecular Biology, Russian Academy of Sciences, 119991 Moscow, Russia; 2Federal Research and Clinical Center of Physical-Chemical Medicine, 119435 Moscow, Russia; 3Institute of Biodesign and Complex System Modeling, Sechenov First Moscow State Medical University, 119146 Moscow, Russia; 4A.V. Topchiev Institute of Petrochemical Synthesis, Russian Academy of Sciences, 119991 Moscow, Russia

**Keywords:** alpha-thrombin, aptamer, G-quadruplex, alpha-deoxyguanosine, peptide-oligonucleotide conjugate, circular dichroism, anticoagulant activity, molecular dynamics

## Abstract

Modification of DNA aptamers is aimed at increasing their thermodynamic stability, and improving affinity and resistance to biodegradation. G-quadruplex DNA aptamers are a family of affinity ligands that form non-canonical DNA assemblies based on a G-tetrads stack. Modification of the quadruplex core is challenging since it can cause complete loss of affinity of the aptamer. On the other hand, increased thermodynamic stability could be a worthy reward. In the current paper, we developed new three- and four-layer modified analogues of the thrombin binding aptamer with high thermal stability, which retain anticoagulant activity against alpha-thrombin. In the modified aptamers, one or two G-tetrads contained non-natural anti-preferred alpha-deoxyguanosines at specific positions. The use of this nucleotide analogue made it possible to control the topology of the modified structures. Due to the presence of non-natural tetrads, we observed some decrease in the anticoagulant activity of the modified aptamers compared to the natural prototype. This negative effect was completely compensated by conjugation of the aptamers with optimized tripeptide sequences.

## 1. Introduction

In recent years, nucleic acid aptamers have attracted increasing attention due to their high potential in molecular therapy and diagnostics. Aptamers are single-stranded DNA or RNA molecules that specifically bind with high affinity to their targets, which may range from small molecules such as adenosine monophosphate to large proteins. Typically, aptamers are evolved in an experimental procedure called SELEX (systematic evolution of ligands via exponential enrichment). This consists of repeated rounds of enrichment of a large DNA or RNA library through competitive binding to a target. The structural basis for the high specificity and affinity of aptamer ligands lies in their ability to form a unique three-dimensional structure that perfectly fits the target. While some aptamers adopt their active conformation primarily through an induced fit mechanism [1,2], others can form the core of their specific spatial structure independently of targets. In particular, the 15-mer DNA aptamer that binds and inhibits alpha-thrombin forms a well-characterized anti-parallel G-quadruplex structure [3,4]. The thrombin binding aptamer (TBA) is a convenient model for studying aptamer–protein interactions, since detailed structural information about the aptamer itself and its complex with the protein is available [5,6,7,8,9]. This circumstance has contributed to numerous attempts to rationally design optimized TBA variants with increased thermodynamic stability, improved affinity, or increased resistance to biodegradation. To date, a large number of TBA-like structures have been proposed, which differ widely in affinity, stability, and resistance to biodegradation [3].

Although multiple strategies have been employed to improve thermodynamic stability or affinity of TBA, adding natural G-quartet planes to an anti-parallel G-quadruplex has never been reported to be a successful approach. It has recently been demonstrated [10] that TBA derivatives with one or two additional G-tetrads do not exhibit anticoagulant properties. A three-layer TBA analogue appeared to adopt a parallel configuration, while a four-layer analogue retained an anti-parallel fold [10,11]. Despite failures with natural three-layer structures, non-natural analogues proved to be more promising. An example of a three-layer TBA analogue with non-natural alpha-deoxyguanosines in two of its three G-tetrads was reported recently by our group [12] (model 1 in Figure 1). An anti-parallel modified G-quadruplex was characterized by notably enhanced thermal stability and retained its anticoagulant properties, which, however, decreased with respect to TBA. Importantly, due to the presence of two G-tetrads with the same polarity, the modified aptamer showed a hybrid-type CD spectrum in both K^+^ and Na^+^ containing buffers. Since alpha-anomers of nucleosides are thought to preferably adopt *anti*-conformation [13,14,15,16], alpha-deoxyguanosine functions as a *syn*-deoxyguanosine mimetic in modified G-quadruplexes (Appendix A). Similar to other nucleoside analogues with a preferred *syn*- or *anti*-conformation [17,18,19], alpha-deoxyguanosine may be used in the design of new G-quadruplex structures with predefined topology and G-quartet polarities.

In the present study, we take advantage of the *anti*-preferred configuration of alpha-deoxyguanosine and develop new non-natural three- and four-layer TBA analogues that form stable anti-parallel G-quadruplex structures and retain their anticoagulant activity. Furthermore, we show that the antithrombotic activity of modified G-quadruplexes can be considerably increased by regioselective conjugation with optimized tripeptide sequences, as recently described [20].

## 2. Results and Discussion

### 2.1. Design of the TBA Analogues

In our recent study [12], we hypothesized that a TBA–thrombin recognition interface is highly sensitive to a proximal G-quartet configuration. Therefore, in the design of new three- and four-layer TBA analogues we tried to preserve a native TBA configuration at the aptamer–protein interface (Figure 1 and Table 1). To this end, we left intact the two G-tetrads proximal to the protein (Q1 and Q2) in models 2–5, thus reproducing the architecture of unmodified TBA. One or two non-natural G-tetrads were added to TBA in four new alpha-thrombin aptamer derivatives (2, 3, 4, and 5), as shown in Figure 1. Based on the published results [13,14,15,16], the conformations of non-natural alpha-deoxyguanosines were presumed as *anti*. In terms of the *syn*–*anti* isomerism of natural deoxyguanosine, modification of the G-quartet with *anti*-alpha-deoxyguanosine at a certain position is equivalent to the *syn* configuration of the unmodified nucleoside at that particular position. Following this rule, we designed modified G-quartets to either retain or reverse the polarity of an adjacent natural quartet. In this way, in TBA analogues 2 and 4 the modified G-tetrads retained the polarity of the natural quartet Q2. However, models 3 and 5 contained modified tetrads opposite in polarity to the natural G-tetrad Q2. In the four-layer modified TBA analogues 4 and 5, two non-natural G-tetrads (Q3 and Q4) were configured to have identical G-quartet polarities by placing two consecutive alpha-deoxyguanosine modifications in the respective strands. TBA analogue 1 represents a variant of the previously studied three-layer model [12], which is characterized by notably improved thermal stability and preserved anticoagulant properties. All TBA analogues in the current study were capped with a tetraethyleneglycol linker and a propargyl functionality at their 3′ end to make possible fluorescent labeling with Cy5 dye via click chemistry (Appendix A).

### 2.2. Binding with Human Alpha-Thrombin

The ability of TBA analogues to bind alpha-thrombin provides important structural information with regard to the organization of the aptamer–protein interface, since the latter is known to require the presence of two TT loops and a specific arrangement of deoxyguanosines in the proximal tetrad. The binding of fluorescently labeled variants of three- and four-layer modified TBA analogues with 2.4 molar excess of protein in 100 mM KCl buffer at 10 °C is shown in Figure 2. For all the studied models, the formation of a fluorescent protein–aptamer complex was observed. A somewhat lower efficiency of binding was detected for variant 5. Overall, this result indicates that all TBA analogues are able to fold into an anti-parallel TBA-like structure and form a specific binding surface.

### 2.3. Thermal Stability and Folding

We further evaluated the thermal stability of G-quadruplex structures formed by modified oligonucleotides in Na^+^ and K^+^ buffer solutions. In our previous study, we found that in 100 mM KCl a variant of model 1 forms an extremely stable structure [12] that does not unfold up to 95 °C. In 100 mM NaCl, we found a *T*_m_ value of 70.4 °C. These findings were confirmed in the present study. Even in a 10 mM K^+^ buffer, we could not observe a transition (Table 1) for analogue 1. The stability of the TBA analogues showed strong correlation with their design. Analogues with identical polarities of adjacent natural and modified tetrads Q2 and Q3 (analogues 2 and 4) turned out to be highly stable. Melting of analogue 4 could only be observed in 10 mM NaCl, and the *T*_m_ value was as high as 87.8 °C. It was found that the G-quadruplex structure formed by aptamer 4 could not be denatured in 7 M urea at room temperature. Gel images showing migration of analogue 4 during electrophoresis in denaturing polyacrylamide gel at different temperatures are given in Appendix A. However, analogues 3 and 5 with opposite polarities of neighboring modified and natural tetrads were characterized by invariably lower *T*_m_ values as compared to variants 2 and 4, respectively. This fact provides evidence that a 5′-(*anti*-dG)-(*anti*-alpha-dG)-3′ step in the G-quadruplex strand induces considerable loss of thermodynamic stability, which can be associated with unfavorable stacking conditions between the modified and unmodified G-quartets Q3 and Q2 in analogues 3 and 5. Interestingly, the presence of two adjacent modified G-tetrads with identical polarity in analogues 1, 4, and 5 was highly beneficial in terms of thermodynamic stability as compared to the structures with a single modified tetrad (2 and 3), regardless of the type of contact between the modified and unmodified G-quartets. Presumably, three- or four-layer stacks of modified tetrads with the same polarity as in 1, 4, and 5 can form highly stable G-quadruplex structures. TBA analogue 2 was the only Q-quadruplex structure that exhibited a two-phase transition in K^+^ containing buffers (Figure 3). This effect was not observed with Na^+^ buffers. Minor low-temperature transitions in 100 and 10 mM K^+^ buffers were observed at 55 and 39 °C, respectively. It is worth noting that the CD spectra of aptamer 2 in 100 mM K^+^ buffer at 20 and 65 °C showed similar patterns (Appendix A), thus providing evidence that the low-temperature transition is not associated with conformational rearrangements.

The CD spectra of TBA analogues generally confirmed the proposed architecture of the new non-natural models (Figure 4). The addition of one modified tetrad Q3 to TBA with polarity identical to the natural tetrad Q2 (model 2) generated a CD pattern that was intermediate between the anti-parallel and typical hybrid type. In the case of analogue 3, the opposite polarities of Q3 and Q2 resulted in preserving the anti-parallel-type CD. A classical hybrid CD pattern was observed for analogues 1 and 4, however, with different contribution from the bands at approximately 260 and 290 nm. The presence of the fourth non-natural tetrad Q4 in aptamer 5 in K^+^ buffer induced minor deviation from the anti-parallel-type CD compared to analogue 3. Importantly, CD patterns in both KCl and NaCl buffers appeared to be very similar for TBA analogues 1, 2, and 4. In the modified G-quadruplex 3, the band at 295 nm acquired a shoulder at 280 nm in Na^+^ buffer. In the case of variant 5, the difference between spectra in K^+^ and Na^+^ buffers became even more obvious. An additional band appeared at approximately 280 nm in 100 mM Na^+^ buffer. Whether the presence of an additional band in the Na^+^ buffer relates to structural rearrangements of the anti-parallel architecture is not clear.

To verify the intramolecular folding of the modified thrombin aptamers, gel-shift assay of G-quadruplexes in K^+^ buffer was carried out using non-denaturing polyacrylamide gel electrophoresis at 10 °C (Figure 5). We used both Cy5-labeled and unlabeled aptamer variants. With one exception, all unlabeled modified G-quadruplexes migrated as a major band in the monomer zone. Aptamer 3 appeared as a major band with a notably lower mobility, which can presumably be associated with the formation of a dimer, bimolecular structure, or unfolded form. The presence of the Cy5 dye at the 3′ end affected the migration patterns of analogues 3 and 5. The former produced smeared bands above the monomer zone. This pattern is consistent with partial unfolding and correlates with the relatively low stability of the G-quadruplex structure formed by aptamer 3. Aptamer quadruplex 5 formed two monomer bands, which most likely resulted from Cy5 dye attachment to a specific functional moiety at the 3′ end. Earlier, we observed the formation of two fluorescent products when using the same type of 3′ clickable tag [12]. Thus, the results of gel-shift assay confirmed the monomolecular folding of TBA analogues 1, 2, 4, and 5. Non-natural G-quadruplex 3 is supposed to fold into a monomolecular structure as well, while showing multiple bands primarily due to partial unfolding. It is not clear whether the major band of unlabeled aptamer 3 is the unfolded form or some other species.

### 2.4. Structural Details of Modified G-Quadruplexes

Molecular dynamics provides an accessible tool for evaluating the structural organization of non-canonical DNA scaffolds. We have applied this technique to characterize five non-natural G-quadruplexes. Starting spatial models of non-natural G-quadruplexes were generated using TBA aptamer models from the previously reported PDB structure 1HAO. We used the previously described strategy to build five non-natural G-quadruplex models [21]. Based on literature reports [13,14,15,16] and the results of CD experiments, *anti*-conformation for non-natural alpha-deoxyguanosines was set in all models. The results of 80-ns MD simulations showed the conformational stability of the modified structures (Appendix A). It was confirmed that the presence of one or two non-natural G-tetrads in three- and four-layer G-quadruplexes is consistent with a TBA-like anti-parallel architecture.

In a series of three-layer G-quadruplexes, variant 2 was identified as the most stable structure by the sum of all contributions to the potential energy (Appendix A). For TBA analogues 4 and 5, we found approximately equal values for the same parameter. The analysis of specific energy contributions (E_q_ + E_VdW_) for each of the three or four tetrads of the studied structures appeared to be more informative (Figure 6). In the three-layer models (1, 2, and 3), tetrad stabilities invariably decreased in the following order: Q3 > Q1 > Q2. Since the terminal modified tetrad Q3 is the least-constrained G-quartet, it assumes the most favorable configuration. The natural tetrad Q1, connected by two TT loops that form an additional TT base pair layer, is somewhat limited in its ability to adopt an advantageous configuration. The location of tetrad Q2 in the middle of the quadruplex core makes it the least stable regardless of its nature. Backbone constraints and different stacking requirements set by the terminal tetrads Q1 and Q3 are the most obvious factors that can reduce the stability of Q2. For both four-layer TBA analogues 4 and 5, the order of G-tetrad stability was as follows: Q4 > Q3 > Q1 > Q2 (Figure 6). The preferred conditions for the tetrad Q3 over Q1 are supposed to be due to the minimum backbone constraints and optimized stacking between Q3 and Q4. The above considerations suggest that the presence of two adjacent identical modified G-tetrads contributes to the overall stability of the quadruplex. However, the contact between modified and unmodified tetrads contributes to the loss of stability. Comparative analysis of aptamers 1, 2, and 3 using the sum (E_q_ + E_VdW_) for all tetrads (Figure 6) clearly shows that the contact between modified and unmodified tetrads with identical polarities (variants 1 and 2) is less disturbing compared to the case of opposite polarities (variant 3). RMSD plots are shown in Appendix A. A snapshot of aptamer 2 from the selected angle of view is shown in Appendix A.

### 2.5. Anticoagulant Activity of Non-Natural TBA Analogues

The ability of non-natural three- and four-layer TBA analogues to bind alpha-thrombin suggests the presence of suitable molecular interface in the aptamer structure. However, the inhibitory characteristics of different structures may vary over a wide range. We examined the anticoagulant activity of new TBA analogues in fibrinogen clotting time tests. Similar to our previous reports [20,22], we used an increase in absorbance at 360 nm due to light scattering during alpha-thrombin-induced fibrinogen polymerization in the presence of an aptamer. The anticoagulant effect was estimated as the time required to reach 50% of the absorbance maximum. Relative t_1/2_ values were calculated by dividing the observed value by the same for TBA. The results of the measurements are shown in Figure 7. Not surprisingly, the modification of the quadruplex core disturbed the recognition interface [12]. A minimal negative effect was observed for analogue 2. Importantly, aptamers 3 and 5, featuring adjacent modified and unmodified G-tetrads with opposite polarity, were the least active in the clotting time test. Nevertheless, the fact that all non-natural variants retained inhibitory activity further supports the design approach based on the use of alpha-guanosine as an *anti*-biased analogue.

### 2.6. Tuning the Activity of TBA Analogues by Conjugation with Optimized Tripeptides

Although most of the studied modified TBA analogues are characterized by very high thermodynamic stability and, presumably, enhanced nuclease resistance due to the presence of non-natural nucleotides, we observed some decline in their anticoagulant activity. Fortunately, in our recent study, we proposed an efficient approach to enhance the anticoagulant activity of TBA [20]. Previously, we have shown that a regioselective conjugation of an optimized tripeptide with a T3 residue at its N3 position greatly increases the anticoagulant activity of the aptamer. The most advanced tripeptide sequence, GLE, boosted the anticoagulant activity of TBA sixfold. Another efficient tripeptide sequence, SLE, increased the anticoagulant activity of TBA by three times. Since the conjugation approach only requires the presence of a canonical TBA recognition interface, in the current paper we applied it to the modified three- and four-layer TBA analogues. The feasibility of this technique was demonstrated with respect to TBA analogues 2 and 4. To this end, we prepared three peptide conjugates 2GLE, 2SLE, and 4GLE. Conjugation with the peptide sequence was carried out at residues T4 in analogue 2 and T5 in analogue 4, which are the functional equivalents of T3 in TBA. In the course of the automated oligonucleotide synthesis, these particular positions were modified with an activated *p*-nitrophenyl thymidine derivative [22]. Post-synthetic conjugation with the respective tripeptides was followed by a standard deprotection procedure yielding crude conjugates. Due to the poor resolution of the complex reaction mixture, we used a combination of reversed-phase HPLC and denaturing gel-electrophoresis at different temperatures to purify peptide–aptamer conjugates. The identity of the conjugates was confirmed by MALDI mass spectrometry.

Predictably, conjugation of TBA analogues 2 and 4 with GLE and SLE tripeptides induced a considerable increase in anticoagulant activity (Figure 7). The relative t_1/2_ values referred to TBA were 2.9 ± 0.6, 5.9 ± 0.9, and 4.1 ± 0.8 for 2SLE, 2GLE, and 4GLE, respectively. Interestingly, the enhancement effect induced by the SLE and GLE peptides in analogue 2 was almost identical in magnitude to that of the previously reported conjugates TBA-SLE and TBA-GLE [20]. In the case of variant 4, the effect of the GLE subunit was apparently weakened by distortions at the aptamer–protein interface due to the presence of two modified tetrads Q3 and Q4 in the quadruplex core.

To characterize the most advanced conjugate 2GLE in terms of binding affinity, we analyzed the interaction between alpha-thrombin and the Cy5-labeled aptamer using a microscale thermophoresis technique. The dissociation constants for Cy5-labeled 2GLE and TBA were determined at an aptamer concentration of 20 nM in 90 mM potassium phosphate buffer (pH 7.5). The aptamer conjugate 2GLE was characterized by improved binding affinity (K_d_ = 2.7 ± 0.5 nM) compared to unmodified TBA (K_d_ = 19.5 ± 1.0 nM).

## 3. Materials and Methods

### 3.1. Materials

Chemical reagents and solvents were purchased from various commercial suppliers and used without further purification. Research-grade human thrombin from plasma for clotting studies was purchased from Renam (Moscow, Russia). Fibrinogen from human plasma was obtained from Sigma-Aldrich (St. Louis, MO, USA). Standard reagents for automated oligonucleotide synthesis were purchased from Glen Research (Sterling, VA, USA). Propargyl controlled pore glass (CPG) was obtained from Lumiprobe (Moscow, Russia). Modified phosphoramidites were purchased from ChemGenes (Wilmington, MA, USA) or were homemade. Synthetic tripeptides were purchased from Cloud-Clone Corp. (Wuhan, China).

### 3.2. Oligonucleotides and Oligonucleotide-Peptide Conjugates

DNA oligomers were synthesized by using an ABI 3400 DNA/RNA synthesizer. Fluorescent aptamers were prepared by click reaction using propargyl-modified oligonucleotides and Sulfo-Cy5 azide (Lumiprobe). Peptide conjugates 2SLE, 2GLE, and 4GLE were prepared as described previously [20]. Purification of TBA analogues, their peptide conjugates, and labeled oligonucleotides was carried out by a combination of reversed-phase HPLC and denaturing gel electrophoresis with control by MALDI mass spectrometry.

### 3.3. Ultraviolet Thermal Denaturation

Absorbance vs. temperature profiles were obtained with a Cary 3500 UV-VIS spectrophotometer (Agilent Technologies, Santa Clara, CA, USA) equipped with a Peltier cell holder. Melting experiments were performed at 295 nm in 10 mM sodium or potassium phosphate (pH 7.5). Concentrations of NaCl or KCl were 0 or 90 mM (the total concentration of Na^+^ or K^+^ ions was 10 or 100 mM). The heating rate was 0.5 °C/min. The melting points were determined from derivative plots of the melting curves. The oligonucleotide concentration was 5 μM. Oligonucleotide samples were slowly annealed in appropriate buffers before analysis.

### 3.4. Circular Dichroism Spectroscopy

Circular dichroism (CD) measurements were performed at 20 or 65 °C by using a Jasco-715 CD spectrometer (JASCO, Easton, MD, USA) with an aptamer concentration of 5 μM in 10 mM sodium or potassium phosphate (pH 7.5). Concentrations of NaCl or KCl were 90 mM. Before measurement, the aptamers were slowly annealed from 95 to 20 °C in the selected buffer.

### 3.5. Binding of Aptamers with Thrombin

Cy5-labeled aptamers were slowly annealed in 10 mM sodium cacodylate (pH 7.2) and 100 mM KCl in D_2_O. Thrombin (12 pmol) was added to 10 μL of an aptamer solution (0.5 μM), and the mixture was incubated for 30 min at 25 C. An analysis of binding was carried out in a 12% native polyacrylamide gel (19:1) at 10 °C in 1 × TBE buffer containing 10 mM KCl. Fluorescent bands of bound and free aptamers were visualized by using a ChemiScope 6000 Series (Clinx Science Instruments, Shanghai, China).

### 3.6. Native Gel-Electrophoresis

Unlabeled aptamers (0.75 nmol) or Cy5-labeled aptamers (5 pmol) were annealed in 10 mM sodium cacodylate (pH 7.2) and 100 mM KCl in D_2_O and analyzed in a 20% native polyacrylamide gel (19:1) at 10 °C in 1 × TBE buffer containing 10 mM KCl. The bands were visualized by UV shadowing or using a ChemiScope 6000 Series (Clinx Science Instruments, Shanghai, China).

### 3.7. Fibrinogen Clotting in the Presence of Aptamers

Human thrombin (50 μL, 10 U/mL) was added to a solution of fibrinogen (2 mg/mL) and aptamer (30 nM) in 1 mL of PBS in a quartz cuvette in a temperature-controlled cuvette holder of a spectrophotometer at 25 °C. Monitoring of the absorbance at 360 nm started immediately after the addition of thrombin and was stopped after the curve reached a plateau. The measurements were carried out in several experimental series. Each series included a blank sample (without aptamer) and TBA control. The clotting curve for each sample was measured at least in duplicate.

### 3.8. Microscale Thermophoresis

The dissociation constants of the aptamer-thrombin complexes were measured with a Monolith NT.115 instrument (NanoTemper Technologies, Munich, Germany) with the Cy5 detection channel. The concentration of the labeled aptamer was 20 nM in 10 mM Tris HCl, 100 mM potassium phosphate (pH 7.5), and 0.5% Tween 20. A standard series of dilutions in the same buffer was used for alpha-thrombin with the highest final concentration in a capillary being 1 mM. The T-jump region was used for automatic K_d_ calculations.

### 3.9. Molecular Dynamics Simulations of TBA Analogues

Essentially, MD studies were performed as described previously [20]. The crystal structure 1HAO was used to build the starting models for modified TBA analogues. The arrangement of the loops matched the same in the crystal structure 1HAO. Modified G-quartets were added to the elements of the starting models using SYBYL X v.2.1 (Certara, Princeton, NJ, USA). The new structures were optimized by the SYBYL X and Powell method with the following settings: partial charges and parameters for interatomic interactions were from the Amber7ff02 force field, a nonbonded cut-off distance was set to 8 Å, a distance-dependent dielectric function was applied, the number of iterations was equal to 500, the simplex method was used in the initial optimization, and the energy gradient convergence criterion of 0.05 kcal/mol/Å was applied.

The model stabilities were verified by MD simulations using Amber 20 software [23]. The influence of the solvent was simulated using the OPC3 water model [24]. Rectangular box and periodic boundary conditions were used in the simulation. The space between the aptamer model and the periodic box wall was at least 15 Å. Potassium ions were used to neutralize the negative charge of the DNA backbone and stabilize the quadruplex structure. The parameters needed for interatomic energy calculation were taken from the force fields OL15 [25,26]. At the beginning of computation, the aptamer models were optimized in two stages. First, the location of the solvent molecules was optimized by using 1000 steps (500 steps of steepest descent followed by 500 steps of conjugate gradient). At this stage, the mobility of all solute atoms was restrained by a force constant of 500 kcal·mol^−1^·Å^−2^. In the second stage, the optimization was performed without restrictions using 2500 steps (1000 steps of steepest descent and 1500 steps of conjugate gradient). Then, gradual heating to 300 K was carried out for 20 ps. To avoid spontaneous fluctuations at this point, weak harmonic restraints were applied with a force constant of 10 kcal·mol^−1^·Å^−2^ for all atoms other than solvent. The SHAKE algorithm was applied to constrain hydrogen-containing bonds, which allowed the use of a 2 fs time step. Scaling of 1–4 nonbonded van der Waals and electrostatic interactions was performed by using the standard Amber values. The cut-off distance for nonbonded interactions was set to 10 Å, and the long-range electrostatics were calculated using the particle mesh Ewald method. MD simulations in the production phase were carried out using constant temperature (T = 300 K) and constant pressure (*p* = 1 atm) over 80 ns. To control the temperature, a Langevin thermostat was used with a collision frequency of 1 ps^−1^. To estimate the distances between tetrads or particular guanines, the centers of mass of the aromatic rings were used. The plots of geometrical parameters and energy of interaction vs. time were smoothed using the moving average method (span = 5).

## 4. Conclusions

Our data on the biophysical, biochemical, and structural characteristics of non-natural three- and four-layer TBA analogues suggest that the use of alpha-deoxyguanosine is a promising approach in the development of new efficient G-quadruplex aptamers. With regard to the whole family of G-quadruplex structures, alpha-deoxyguanosine can be considered in line with other modified or natural nucleotides that regulate assembly of these non-canonical structures through a preferential *syn*–*anti* conformation mechanism. The addition of modified tetrads to TBA provides a considerable increase in the thermodynamic stability of the resulting structures. The payoff for this gain is certain distortions in the quadruplex core that affect the aptamer–protein interface. Fortunately, a moderate loss of anticoagulant activity may be completely abrogated by creating an extended recognition interface through conjugation to an optimized tripeptide subunit. Modified aptamer variants 2 and 4 with the GLE peptide subunit have proven to be promising candidates for advanced anticoagulant DNA aptamers.

## Figures and Tables

**Figure 1 ijms-24-08406-f001:**
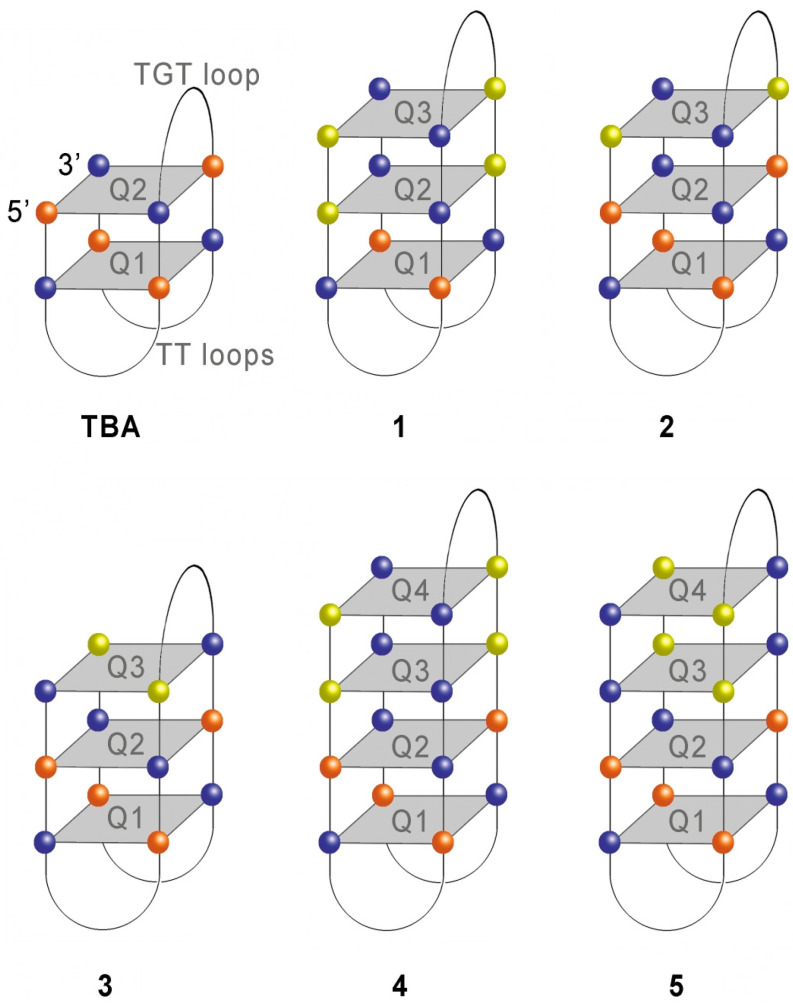
Schematic representation of the TBA architecture and proposed structures for modified models 1–5. The colored spheres designate *syn*-dG (orange), *anti*-dG (blue), and *anti*-alpha-dG (yellow).

**Figure 2 ijms-24-08406-f002:**
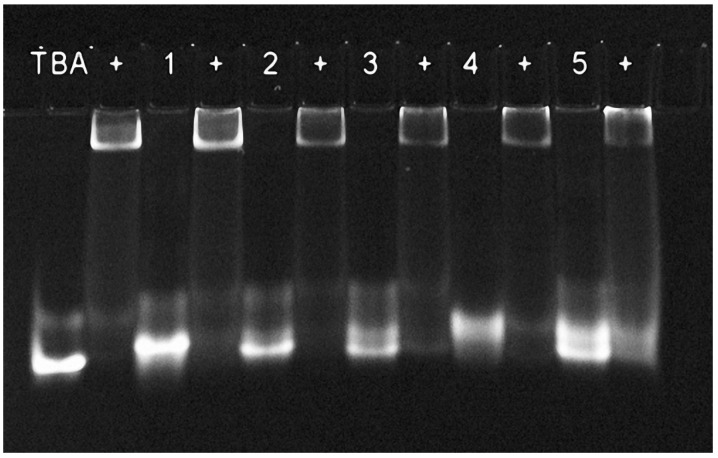
Analysis of the binding of Cy5-labeled aptamers to alpha-thrombin by electrophoresis in a 12% polyacrylamide gel at 10 °C in K^+^ containing buffer. Each lane contained 5 pmol of the aptamer, either alone or in the presence of 2.4 equivalents of thrombin. Symbol (+) indicates added thrombin. All five modified G-quadruplexes bind thrombin, which confirms anti-parallel architecture and the presence of a specific recognition interface in the aptamer structure.

**Figure 3 ijms-24-08406-f003:**
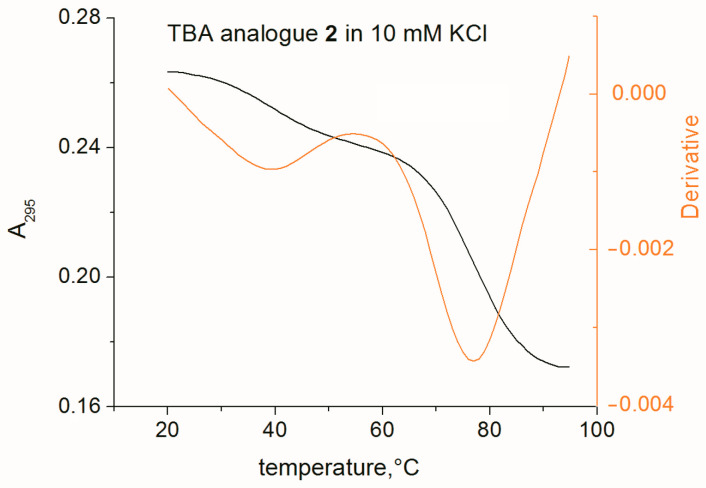
A two-phase melting profile of TBA analogue 2 in the presence of 10 mM K^+^ (black), and a first-derivative plot of the melting curve (orange).

**Figure 4 ijms-24-08406-f004:**
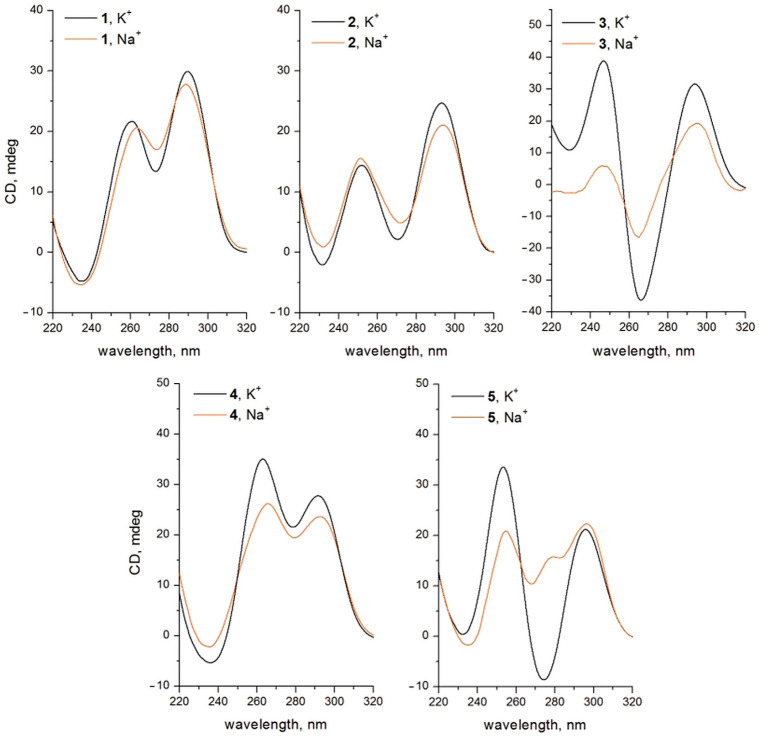
CD spectra of modified TBA analogues at 20 °C in 100 mM K^+^ and Na^+^ containing buffers.

**Figure 5 ijms-24-08406-f005:**
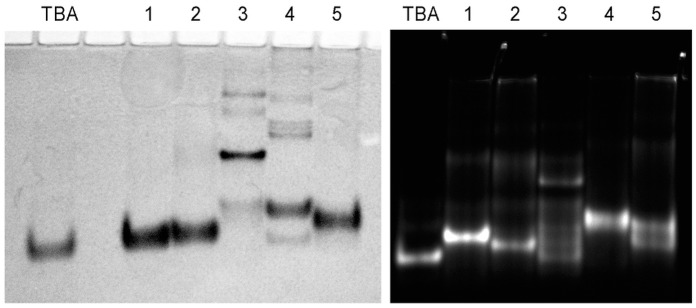
Native electrophoresis of the modified aptamers in 20% polyacrylamide gel in the presence of 10 mM KCl at 10 °C. (**Left panel**): unlabeled aptamers. (**Right panel**): Cy5-labeled aptamers. Band positions confirm monomolecular folding for models 1, 2, 4, and 5. The low-stability modified aptamer 3 exhibits the ability to form slow migrating species.

**Figure 6 ijms-24-08406-f006:**
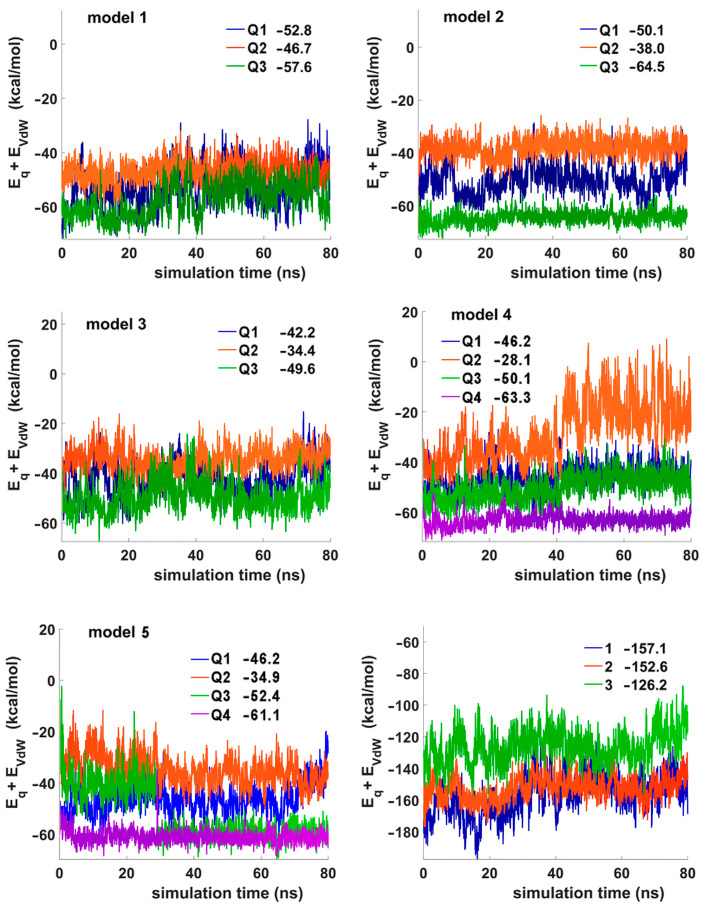
The sum of electrostatic and Van der Waals energy contributions for each G-quartet in TBA analogues 1–5, and the sum of electrostatic and Van der Waals energy contributions for all tetrads in analogues 1–3 (last panel). The average values are shown separately in each panel.

**Figure 7 ijms-24-08406-f007:**
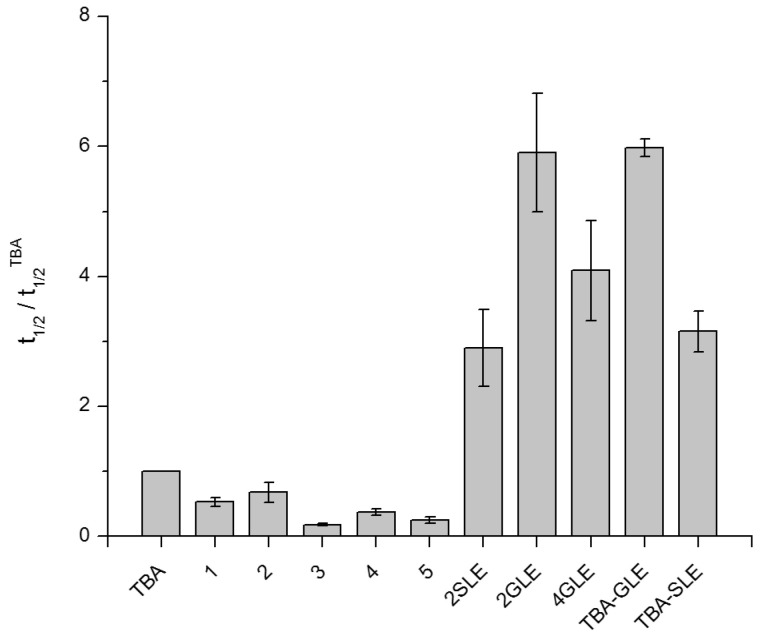
The results of clotting time tests for TBA, modified aptamers 1–5, and aptamer–peptide conjugates. Data for TBA-GLE and TBA-SLE were adapted from [20].

**Table 1 ijms-24-08406-t001:** Modified three- and four-layer TBA analogues and their thermal stabilities.

ID	Sequence ^1^	*T*_m_, °C ^2^			
		100 mM K^+^	10 mM K^+^	100 mM Na^+^	10 mM Na^+^
1	5′-ggGTTGGGTGTggGTTGGG	n.d. ^3^	n.d.	74.5	56.4
2	5′-gGGTTGGGTGTgGGTTGGG	>90 ^4^	76.9	61.3	45.1
3	5′-GGGTTGGgTGTGGGTTGGg	59.1	41.8	44.2	26.5
4	5′-ggGGTTGGGGTGTggGGTTGGGG	n.d.	n.d.	n.d.	87.8
5	5′-GGGGTTGGggTGTGGGGTTGGgg	>90	79.6	65.9	43.0

^1^ g = α-dG; all oligonucleotides have 3′ tetraethylene glycol and propargyl tag (Appendix A). ^2^
*T*_m_ values are determined from first-derivative plots; Δ*T*_m_ = ±0.5 °C; ^3^ transition was not detected up to 95 °C; ^4^ only the beginning of the transition was observed. *T*_m_ values of around 20 and 50 °C have been reported in the literature for unmodified TBA in 100 mM Na^+^ or K^+^ buffers, respectively [3].

## Data Availability

Not applicable.

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
