# Peer review of "Alpha-Deoxyguanosine to Reshape the Alpha-Thrombin Binding Aptamer"

_ijms, 2023, doi:10.3390/ijms24098406_

Round 1

Reviewer 1 Report

Summary:

The manuscript describes synthesis and evaluation of five TBA-based DNA aptamers containing modified bases (specifically, alpha-deoxyguanosine) at different positions in the quadruplex. The aptamers are characterized by measurement of thermal stability and binding affinity to thrombin, as well as evaluation of their ability to inhibit clotting. All modified aptamers were less effective at preventing clotting compared to unmodified TBA. However, when conjugated to tripeptides, the aptamers became much more effective. Binding affinities of the conjugated aptamers were fairly strong, in the low nM range.

General comments:

The authors present a well-written and concise report of their study. There are a number of points in the Results and Discussion section that would benefit from some clarity:

1. Table 1 shows the thermal stability of the aptamers 1-5 in Na+ and K+ containing buffers. How do these compare to the native TBA stability? Also, what is the significance of comparing stability in K+ versus Na+?

2. The term polarity is used to describe the configuration of the tetrads. I'm a little confused about the meaning - does polarity refer to syn vs. anti, or alpha vs. beta anomer?

3. In Figure 3, the melting curve is displayed in a way that I'm not accustomed to seeing. Instead of A/Amin being displayed on the y-axis, it might be easier to interpret if the y-axis was simply A295.

4. Looking at the native electrophoresis data in Figure 5, the mobility of aptamer 3 is clearly decreased, which is interpreted by the authors as potential formation of a dimer. However, according to Table 1, aptamer 3 is also far less stable than the others. The electrophoresis is performed at a temperature that is lower than the measured Tm in 10 mM K+, so it's probably unlikely that it is becoming fully unfolded. Perhaps there is some heterogeneity in folding, causing a number of different structures; or maybe the aptamer is partially unfolded? That could possibly explain both the reduced mobility and the smearing seen in the Cy-5 labeled protein.

5. I'm having a hard time understanding the significance of the molecular dynamics data presented in Figure 6. In most cases, the data seem very noisy and aside from the Q2 data in model 4, don't seem to differ much from one another. My suggestion would be to move the plotted data to the supplementary information and report the average values (with standard error) as a table in the text. Regardless of where the graphical data is shown, the data shown in each panel should not be in colors that are so similar because it is difficult to distinguish them from one another.

6. For the clotting time tests: I would suggest adding the data from the bar graph on Figure 8 to Figure 7, so they can all be compared more easily. The actual clotting curves can also probably be moved to the supplementary information, or removed altogether. Also, was the control for the conjugated aptamers (TBA conjugated to GLE or SLE) done? it would make more sense to compare the modified aptamer conjugates to that, rather than to unconjugated TBA.

7. Again, the calculated Kds are compared between 2GLE and unmodified TBA; this doesn't seem like a fair comparison. I would like to see binding data for a conjugated TBA. On a related note, if the instrumentation for microscale thermophoresis is available, why use electrophoretic mobility shift assays (as presented in figure 2)? That way, it is possible to get comparable Kds for all of the unconjugated aptamers rather than just a qualitative assessment from the gel. Also,

Minor changes:

1. Lines 66-68: this is unclear - if alpha-anomers adopt primarily anti-positions, why would it function as a syn-deoxyguanosine mimetic?

2. Likewise, in lines 96-99 - the sentence that starts with "In terms..." is difficult to follow. Is there a figure or diagram that might help? This is probably intuitive to someone who is familiar with nucleoside chemistry, but since this journal is intended for a broader readership, it would be helpful if this could be explained a little better.

3. In Table 1, n.d. is explained in the caption as having a transition that could not be detected up to 95 deg C. There are also two entries that give the Tm as >90 deg C. This is picky, but I would change them all to read n.d. or >90.

Author Response

The authors would like to thank the reviewer for careful reading and detailed analysis of the text.

  1. We provided the information about the thermal stability of unmodified TBA in the footnote to Table 1. Since there are numerous reports on TBA stability, we placed a reference to a review paper that combined all available data. With regard to significance of the Tm values measured in different conditions, it makes sense for two reasons. First, the stability of G-quadruplexes is usually higher in K+ buffers. Therefore, using different buffers we can compare structures with very high or very low stabilities. In our case, aptamers 1 and 4 stabilities could be determined in Na+ buffers only. Second, our data on thermal stabilities are complemented by CD experiments, which allow structural analysis of aptamer folding. Since the quadruplex topology in some cases is dependent on the counter ion, it was reasonable to verify such a possibility.
  2. The term of G-tetrad polarity relates to the pattern of hydrogen bonding in the particular G-quartet and depends on the relative strand polarities and syn-anti configuration of guanines. This term is commonly accepted and rarely described in details in a typical research paper. There is a good overview of G-quadruplex structural features here: https://ericlarg4.github.io/Distill_section/docs/guideline.html;
  3. We changed axis in the figure.
  4. Thank you for this comment. After discussion between coauthors, we concluded that unfolded state seems to be the most probable form for the slow-migrating band of 3. Nevertheless, at this point, we cannot reliably confirm this statement. We modified this part of the text.
  5. We agree that the color scheme had to be improved. This issue was corrected. With regard to significance of the MD data, it provided the only tool to understand the detailed structural organization of the new G-quadruplex scaffolds and to compare energetics between different structures and different elements within a single structure. A large part of the data is given in Supplementary file. Even larger part is not included into the paper. We prefer to show graphical data on E(VdW)+E(q) in the paper, since it illustrates the text in section 2.4. Variations in E values (between quartets and within a single quartet) through the course of simulation reflect the stability of structural elements with respect to each other. Therefore, visual presentation is very important. The term standard error is not applicable since the values are calculated, and not measured experimentally.
  6. We combined the figures 7 and 8 as proposed by the reviewer. Also, we added reference data for TBA-GLE and TBA-SLE from paper [20]. Clotting curves were removed.
  7. In fact, it is Kd data for TBA and conjugate, that makes sense to compare. Improved Kd values, as well as increased clotting times, result from the extended recognition interface. Comparison between two almost identical structural subunits (in 2-GLE and TBA-GLE) would result in very similar Kd values, as observed for the clotting time values. Since we had limited access to the Nanotemper equipment and due to timing considerations, we selected only the most advanced variant and TBA reference for estimating Kd values. We used the gel-electrophoresis technique to inspect visually band shifts in the binding assays. This possibility is not available in the thermophoresis method, as it provides only binding curve.

Minor changes:

  1. The illustration of how alpha-dG(anti) mimics natural dG(syn) is given in the new figure S1. Also, the consideration that, in mixed DNA duplex (alpha+beta strands), both strands are parallel may be helpful.
  2. See above.
  3. We added comments to Table 1. N.d.=no transition was observed up to 95C; >90=We observed only the beginning of a transition.

Reviewer 2 Report

Although this manuscript provides certain level of scientific data, the manuscript itself was not well prepared in well-understandable ways. Please think about the following points.

1) Figures and figure captions are not well described in understandable ways (especially electrophoresis results). Figures have to be prepared. Without reading carefully main texts, the figures have to be prepared in well understandable.

2) Illustration on structural changes have to be represented. Otherwise, main point of this research cannot be well understood.

Author Response

The authors would like to thank the reviewer for valuable advice. To address the reviewer’s concerns, we modified legends for Figures 2, 3, and 5, modified Figure 7, and added new Figures S1 and S10.

Reviewer 3 Report

The manuscript by Kolganova et al on artificial aptamer construct with increased thermal stability and anti-coagulative property is an interesting read. I have a few minor comments to make pertaining to the computational sections. 

Firstly, please do not refer to your previous publication to describe a method. The reader should be able to do and/or reproduce reported  simulations without searching another article. So please describe methodology used in this manuscript in relevant places.

The last panel of Figure 6, does not require such a large scale. Using a smaller scale has an advantage of showing differences much clearly.

The MD force fields do not carry any specific terms and nor they are parameterized for stacking interactions. Discussions based on parallel stackings from MD simulation energetics are to be taken with a lot of salt. If the authors want to comment of the stacking interactions as the possible stability marker, they may want to check the dihedral distribution between select rings.     

Author Response

  1. It is a common practice in other fields of science. To address the reviewer’s concern, we placed a detailed description in Methods as requested.
  2. Thank you, this was corrected.
  3. Indeed, the MD force fields do not carry any specific terms and nor they are parameterized for stacking interactions. Fortunately, stacking interactions are well described by the van der Waals potential and charge electrostatics in AMBER force fields. Please, have a look at: DOI 10.1002/bip.22322. Another consideration is that stacking is a widely used concept in structural biology, which would be very useful in describing MD results for non-specialists.

Round 2

Reviewer 1 Report

All of my concerns have been adequately addressed. Best wishes!

Reviewer 2 Report

It looks OK.